# Diving into a holographic superconductor

Sean A. Hartnoll[1], Gary T. Horowitz[2], Jorrit Kruthoff[1] and Jorge E. Santos[3,4]

[1] *Department of Physics, Stanford University, Stanford, CA 94305-4060, USA*

[2] *Department of Physics, University of California, Santa Barbara, CA 93106, USA*

[3] *DAMTP, University of Cambridge, Wilberforce Road, Cambridge CB3 0WA, UK*

[4] *Institute for Advanced Study, Princeton, NJ 08540, USA*

January 18, 2021

### Abstract

Charged black holes in anti-de Sitter space become unstable to forming charged scalar hair at low temperatures $T < T_c$. This phenomenon is a holographic realization of superconductivity. We look inside the horizon of these holographic superconductors and find intricate dynamical behavior. The spacetime ends at a spacelike Kasner singularity, and there is no Cauchy horizon. Before reaching the singularity, there are several intermediate regimes which we study both analytically and numerically. These include strong Josephson oscillations in the condensate and possible 'Kasner inversions' in which after many e-folds of expansion, the Einstein-Rosen bridge contracts towards the singularity. Due to the Josephson oscillations, the number of Kasner inversions depends very sensitively on $T$, and diverges at a discrete set of temperatures $\{T_n\}$ that accumulate at $T_c$. Near these $T_n$, the final Kasner exponent exhibits fractal-like behavior.

# 1 Introduction

Over a decade ago, a holographic realization of superconductivity was found [1–3]. Charged black holes, such as the Reissner-Nordström anti-de Sitter (RN AdS) solution, dually describe nonzero density states of matter. Such black holes were shown to become unstable to forming charged scalar hair at low temperatures $T < T_{\rm c}$. The hairy solutions spontaneously break the $U(1)$ symmetry of the theory, leading to the physics of superconductors and superfluids. Since that time, holographic superconductors have been extensively studied [4, 5]. However, to the best of our knowledge, the effect of the charged condensate on the black hole interior, beyond the horizon, has not been systematically studied. This paper aims to fill this gap.

The well-studied exterior of a holographic superconductor is relatively simple: a 'lump' of scalar field is localized close to the horizon, held in place by the gravitational pull towards the interior of AdS combined with electrostatic repulsion from the horizon. We will find that, in contrast, the interior exhibits intricate dynamics which can be divided up into several different epochs. We will characterize these epochs both analytically and numerically. We will also prove that there cannot be a smooth Cauchy horizon. Instead, the solutions approach a spacelike singularity at late interior time.

Before describing these different regimes, we should clarify our motivation. Of course the reason there has been little interest in the solution behind the horizon is that it is not clear how this region of the geometry is reflected in the dual symmetry-broken phase of matter.

While certain entanglement in the thermofield double state is captured by a transhorizon extremal surface [6], these surfaces do not probe far enough beyond the horizon to see the regimes that we will describe (cf. [7]). In the spirit of the recent works [8,9], we hope that the existence of nontrivial classical dynamics behind the horizon will help to motivate and guide the search for a more powerful holographic understanding of the black hole interior.

Just below the critical temperature, the scalar field is uniformly small everywhere outside the horizon. Inside the horizon, we will see that the dynamics cleanly separates into epochs that we call the collapse of the Einstein-Rosen (ER) bridge, Josephson oscillations, Kasner, and in some cases, Kasner inversions. We now explain this terminology. The solution remains close to RN AdS until one approaches the inner horizon. At that point the direction along the Einstein-Rosen bridge shrinks very rapidly while the two transverse directions are essentially unchanged. This is the collapse of the Einstein-Rosen bridge, similar to that seen previously for a neutral scalar field [9]. Following this, we find that the scalar field undergoes rapid oscillations which are analogous to Josephson oscillations in a superconductor. When these oscillations end, the solution resembles the Kasner solution, which is a homogeneous, anisotropic cosmology where the metric components are all power laws and the scalar field is logarithmic [10].

Our Kasner solutions have a single free exponent $p_t$ which takes values $-1/3 \leq p_t \leq 1$. The value of $p_t$ after the oscillations depends on temperature. When $p_t$ is positive it remains constant all the way to the singularity. This corresponds to $g_{tt}$ continuing to decrease to zero. However if $-1/3 < p_t < 0$, $g_{tt}$ starts growing and after many e-folds of expansion there is a transition to another Kasner regime. Most of the negative exponents get mapped to positive values, so $g_{tt}$ again decreases to the singularity. We call this phenomenon a 'Kasner inversion'. However a small neighborhood of $p_t = -1/3$ is mapped to new negative values. This occurs because in these cases the inversion is so sudden that additional Josephson oscillations in the scalar field are induced. In such cases the expansive dynamics then continues for many e-folds until it reaches a second Kasner inversion where the process repeats. Each time, the range of temperatures for which $p_t$ remains negative becomes smaller and smaller. For a discrete set of temperatures $\{T_n\}$, there are an infinite number of Kasner inversions, making the final $p_t$ extremely sensitive to the temperature. These special $T_n$ accumulate at $T_c$ showing that the onset of superconductivity is accompanied by extremely intricate interior dynamics.[1]

---

[1]The sequence of Kasner regimes resembles the well-known BKL approach to spacelike singularities [11]. However, our results do not follow from previous analyses: The oscillations induced by the charge of the scalar field are essential to explain how the number of Kasner epochs depends sensitively on the black hole temperature.

We will present approximate analytic solutions for each of the different interior epochs, and confirm their accuracy by matching to numerical solutions. These include the collapse of the ER bridge given by (17) and Fig. 3, the scalar oscillations given by (21) and Fig. 4, and the Kasner inversion given by (32) and Fig. 9. Away from $T_c$ the interior solution typically has less structure, but there are still some critical temperatures where there are an infinite number of Kasner inversions. We thus obtain a fairly complete picture of the classical solution inside a holographic superconductor.

To obtain our approximate analytic solutions, we first study the numerical solutions to see which terms in the field equations are small in a given epoch. We then drop those terms and solve the remaining equations analytically. The self-consistency of this procedure is established by checking that the dropped terms are small on the analytic solution (and parametrically so in certain limits). Finally, we verify that the numerical solutions indeed match our analytic approximations in each epoch. Although we cannot justify a priori why the terms we drop are negligible, the analytic approximations provide considerable insight as they explain complicated behavior in terms of a few parameters.

As usual, one expects this interior solution will break down near the singularity and require stringy or quantum corrections. Large curvatures can also arise in limiting cases where the ER bridge collapse or Kasner inversions become arbitrarily sudden. In holography we can control when these corrections become important by taking the two parameters of the dual gauge theory, $N$ and the coupling constant $\lambda$, very large. This will often be enough to ensure that such corrections remain small until we are late into the final Kasner epoch or very close a limiting Kasner inversion.

## 2   Review of holographic superconductors

A minimal holographic superconductor is described by gravity coupled to a Maxwell field and a charged, massive scalar field with action [1–3]

$$S = \int d^4x \sqrt{g} \left[ R + 6 - \frac{1}{4}F^2 - g^{ab}(\partial_a \phi - iqA_a\phi)(\partial_b \phi + iqA_b\phi) - m^2\phi^2 \right] . \qquad (1)$$

We have set the AdS radius and gravitational coupling to one. The Maxwell field is dual to the current of a global $U(1)$ symmetry in the field theory. The scalar field $\phi$ is dual to an operator $\mathcal{O}$ with charge $q$ under this global symmetry, and scaling dimension $\Delta = \frac{3}{2} + \sqrt{\frac{9}{4} + m^2}$.

We will be interested in planar black hole solutions of the form

$$ds^2 = \frac{1}{z^2} \left( -f(z)e^{-\chi(z)}dt^2 + \frac{dz^2}{f(z)} + dx^2 + dy^2 \right) , \qquad (2)$$

The AdS boundary is at $z = 0$ and the singularity will be at $z \to \infty$. At a horizon, $f(z_{\mathcal{H}}) = 0$. The horizon defines the temperature

$$T = \frac{1}{4\pi} |f'(z_{\mathcal{H}})| e^{-\chi(z_{\mathcal{H}})/2} \,. \tag{3}$$

The scalar field and scalar potential take the form

$$\phi = \phi(z) \,, \qquad A = \Phi(z) \, \mathrm{d}t \,. \tag{4}$$

The defining feature of a holographic superconductor is that the operator $\mathcal{O}$ condenses below some critical temperature $T_{\mathrm{c}}$, spontaneously breaking the global symmetry. For the model (1), $T_{\mathrm{c}}$ depends on $q$ and $\Delta$ as shown in [12]. Below the critical temperature in the bulk, the scalar field develops a nonzero normalizable falloff at the boundary, without a source, so that as $z \to 0$

$$\phi \to \phi_{(1)} z^{\Delta} \,, \tag{5}$$

with $\phi_{(1)} \propto \langle \mathcal{O} \rangle$. The remaining radial functions should have the leading asymptotic behavior: $f \to 1 \,, \chi \to 0 \,, \Phi \to \mu$. This behavior fixes the normalization of time on the boundary as well as the chemical potential $\mu$.

The equations of motion are

$$z^2 e^{-\chi/2} \left( e^{\chi/2} \Phi' \right)' = \frac{2q^2 \phi^2}{f} \Phi \,, \tag{6}$$

$$z^2 e^{\chi/2} \left( \frac{e^{-\chi/2} f \phi'}{z^2} \right)' = \left( \frac{m^2}{z^2} - \frac{q^2 e^{\chi} \Phi^2}{f} \right) \phi \,, \tag{7}$$

$$\frac{\chi'}{z} = \frac{q^2 e^{\chi}}{f^2} \phi^2 \Phi^2 + (\phi')^2 \,, \tag{8}$$

$$4 \, e^{\chi/2} z^4 \left( \frac{e^{-\chi/2} f}{z^3} \right)' = 2 \, m^2 \, \phi^2 + z^4 e^{\chi} (\Phi')^2 - 12 \,. \tag{9}$$

The equations of motion also fix the phase of $\phi$ to be constant, so we can choose $\phi$ to be real. Previous works have solved these equations in the black hole exterior $z > z_{\mathcal{H}}$. To continue behind the horizon it is simple to go to ingoing coordinates [9], and the equations of motion for $f, \chi, \phi, \Phi$ do not change.

## 3 Proof of no inner horizon

In the absence of the charged condensate, *e.g.* $\phi_{(1)} = 0$ for $T > T_{\mathrm{c}}$, the bulk solution is the charged RN AdS black hole. The interior of RN AdS has an inner Cauchy horizon and a timelike singularity. In recent work we have shown that neutral scalar fields generically

destroy the inner horizon and lead to a spacelike singularity [9]. Here we establish a stronger result for charged scalar fields: smooth Cauchy horizons never form.

An important new ingredient for the case of a charged scalar field is that several terms in the equations of motion $(6) - (9)$ have inverse factors of $f$ in them. At any horizon, $f$ vanishes. It is easily seen that if the variables are real analytic at the horizon, *i.e.* admit a power series expansion, then $\phi$ or $\Phi$ must also vanish at the horizon. The series expansion at the horizon furthermore shows that $\phi$ can only vanish on the horizon if it vanishes everywhere. Therefore, in the presence of a nonzero condensate $\phi$, the potential $\Phi$ must vanish on all horizons. We now show that it is not possible for $\Phi$ to simultaneously vanish on both an inner and an outer horizon.

On our ansatz for the fields, the action $(1)$ is invariant under $z \to \lambda z, \chi \to \chi - 6 \log \lambda, \Phi \to \lambda^2 \Phi$, with $f$ and $\phi$ not changing. The associated conserved quantity $Q$ is [13]

$$Q \equiv \frac{e^{\chi/2}}{z^2} \left( e^{-\chi} f \right)' - e^{\chi/2} \Phi' \Phi \,. \tag{10}$$

The fact that $Q' = 0$ follows from the equations of motion. Since $Q$ is constant, its value must agree on horizons, where $f = \Phi = 0$ (with a nonzero $\phi$). It follows that if there were two horizons, at $z_{\mathcal{H}}$ and $z_{\mathcal{I}}$, we would need

$$\left. \frac{e^{-\chi/2} f'}{z^2} \right|_{z_{\mathcal{H}}} = \left. \frac{e^{-\chi/2} f'}{z^2} \right|_{z_{\mathcal{I}}} \,. \tag{11}$$

However, this is impossible because $f'$ is negative on an outer horizon and positive on an inner horizon. Therefore, in the absence of an inner horizon, we can anticipate that the interior geometry will end at a spacelike singularity.

*Note added:* This proof has been extended to spherical horizons in the recent paper [14].

## 4  Dynamical epochs inside the horizon

Beyond the horizon of the holographic superconductor, the radial coordinate $z$ is timelike. In this section we will describe several distinct dynamical regimes that can occur as the interior geometry evolves from the horizon to the spacelike singularity. These are (i) the collapse of the Einstein-Rosen bridge, (ii) Josephson oscillations of the condensate and (iii) a Kasner cosmology, sometimes with transitions that change the Kasner exponents. We will give analytic descriptions of each of these regimes in certain limits, that match numerical results. Fig. 1 gives an overview of these different dynamical epochs, while Fig. 2 shows a zoom into the collapse and oscillation regimes.

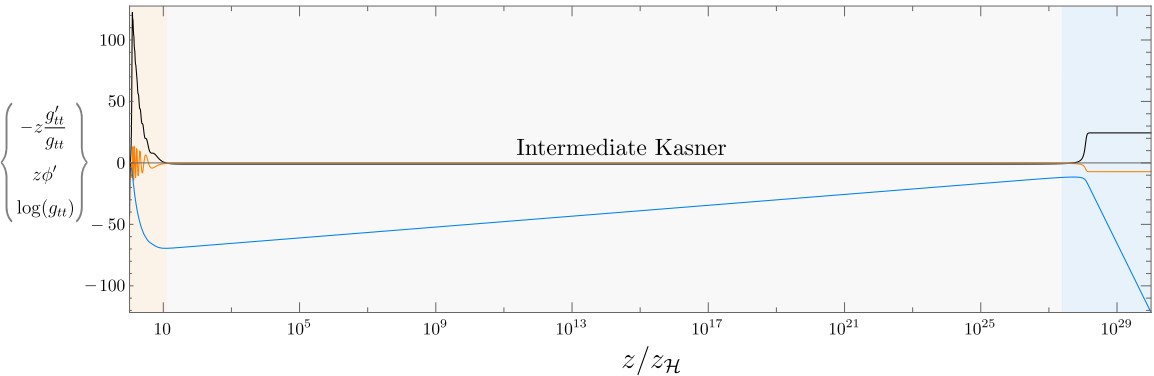

Figure 1: Journey through the inside of a holographic superconductor. At a value of $z$ close to the inner horizon of AdS RN, $zg'_{tt}/g_{tt}$ (black) experiences a large kick at which $g_{tt}$ (blue) becomes very small. We call this the collapse of the Einstein-Rosen bridge. Immediately afterwards, the scalar field $\phi$ (orange) goes through a series of Josephson oscillations, imprinting a series of short steps on $zg'_{tt}/g_{tt}$. This all occurs at relatively small $z/z_{\mathcal{H}}$ (orange shaded region, shown also in Fig. 2). The oscillations settle down to an intermediate Kasner regime with exponent $p_t^{\text{int}}$. This Kasner epoch lasts for an exponentially long range of $z/z_{\mathcal{H}}$ (but short proper time) before receiving yet another kick (whenever $p_t^{\text{int}} < 0$) after which the interior has a final Kasner exponent with $p_t = -p_t^{\text{int}}/(2p_t^{\text{int}} + 1) > 0$ (blue shaded region). The scalar field derivative inverts: $z\phi' \to 2/(z\phi')$. Here $q = 1$, $m^2 = -2$ and $T/T_c = 0.986$.

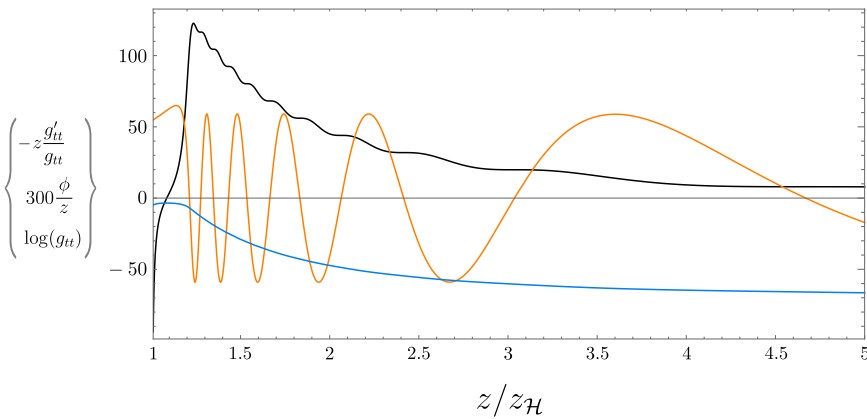

Figure 2: The shaded orange region of Fig. 1. The same colors have been used, but $300\,\phi/z$ is plotted instead of $z\phi'$ to highlight the oscillations and their amplitude. The imprint of the oscillations on the metric are clearly visible. Here $q = 1$, $m^2 = -2$ and $T/T_c = 0.986$.

## 4.1 Simplified interior equations of motion

The regimes beyond the horizon are most cleanly identified in the limit where the scalar field is small. This will be the case, for example, just below the critical temperature $T_c$. Figs. 1 and 2 are in this limit, as is most of our analytic discussion that follows. In §4.4 we show how the Einstein-Rosen bridge collapse and Josephson oscillations become less distinct at lower temperatures. At temperatures close to $T_c$ the effect of the scalar field is small until the solution comes close to the would-be inner horizon of RN AdS. At that point there are strong nonlinearities that lead to novel dynamics. One can verify numerically (and validate a posteriori) that by the time the interesting dynamics kicks in, the following terms in the equations of motion can be dropped: the mass terms of the scalar field and the charge term in the Maxwell equation.[2] The equations (6) — (9) then become

$$\Phi' = E_o\, e^{-\chi/2}\,, \tag{12}$$

$$\frac{e^{-\chi/2} f}{z^2}\left(\frac{e^{-\chi/2} f \phi'}{z^2}\right)' = -\frac{q^2 \Phi^2}{z^4}\phi\,, \tag{13}$$

$$\frac{\chi'}{z} = \frac{q^2 e^{\chi}}{f^2}\phi^2 \Phi^2 + (\phi')^2\,, \tag{14}$$

$$\left(\frac{e^{-\chi/2} f}{z^3}\right)' = \frac{1}{4}\left(E_o^2 - \frac{12}{z^4}\right) e^{-\chi/2}\,. \tag{15}$$

Here $E_o$ is the constant electric field in this regime. The electric field is in the spacelike $t$ direction, while $\Phi$ should be thought of as a component of the vector potential inside the horizon, with the $z$ coordinate being 'time'. The term on the right hand side of the Maxwell equation (6) is therefore a Josephson electric current in the interior, due to the condensate $\phi$ and vector potential $\Phi$. The Josephson current is dropped in (12) because it does not backreact significantly on the electric field in the regimes we are about to describe.

---

[2]It is not simply $\phi$ being small close to $T_c$ that allows terms to be dropped. Rather, we will see in the solutions below how nonlinear dynamics can result in certain quantities becoming exponentially large compared to others. In §4.5 this will also occur away from $T_c$. Substituting these behaviors into (6) — (9) reveals that the mass terms are negligible. We will be dropping additional terms as we proceed further beyond the horizon. As noted in the introduction, the behaviors that allow us to do this are not obvious a priori. Our methodology throughout has been to find self-consistent analytic approximations guided by numerical exploration.

## 4.2 Collapse of the Einstein-Rosen bridge

The physics here is similar to that discussed recently for a neutral scalar field in [9]. A small nonzero scalar triggers an instability of the would-be inner Cauchy horizon. The instability is stronger for small values of the scalar, indicating the nonlinear nature of the dynamics. The essential phenomenon is that as $g_{tt}$ approaches its would-be zero at the Cauchy horizon, it suddenly undergoes a very rapid collapse to become exponentially small. In [9] we called this the collapse of the Einstein-Rosen bridge.

As in [9], for vanishingly small scalar field the instability becomes so fast that the $z$ coordinate can be kept essentially fixed. Let $z = z_\star + \delta z$, so that $f, \chi, \phi, \Phi$ are now functions of $\delta z$ while any explicit factors of $z$ in the equations are set to $z_\star$. The constant $z_\star$ will be close to the inner horizon of RN-AdS. Furthermore, in this limit the potential $\Phi$ is large compared to its derivative in this regime (we will verify this after the fact). Thus in (13) and (14) we can set $\Phi = \Phi_o$, a constant (it is important to keep the $E_o$ term in (15) however). With these approximations, equation (13) can be solved explicitly as

$$\phi = \phi_o \cos\left( q\Phi_o \int_{z_\star}^z \frac{e^{\chi/2} dz}{f} + \varphi_o \right) . \tag{16}$$

Here $\phi_o$ and $\varphi_o$ are constants. Note that if $f$ develops a zero then one has the expected logarithmic oscillations close to the inner horizon seen in studies of the linear instability of this horizon.

Perhaps remarkably, the scalar field oscillations in (16) drop out of the remaining equations for $f$ and $\chi$. This is because $q^2\phi^2\Phi_o^2 + e^{-\chi}f^2(\phi')^2 = q^2\phi_o^2\Phi_o^2$ in (14). These equations are then seen to be identical to those of a neutral scalar field that were solved in [9]. In particular, the metric component $g_{tt}$ is found to be given by

$$c_1^2 \log(g_{tt}) + g_{tt} = -c_2^2(z - z_o), \qquad c_1^2 = \frac{2q^2\phi_o^2\Phi_o^2}{z_\star^4 E_o^2 - 12}, \tag{17}$$

and $c_2 > 0$ and $z_o$ are further constants of integration. For $z < z_o$, $g_{tt} \propto (z_o - z)$ is linearly vanishing, as in the approach to an inner horizon, but for $z > z_o$ we see that instead of vanishing or changing sign, $g_{tt} \propto e^{-(c_2/c_1)^2(z-z_o)}$ is nonzero but exponentially small. This collapse occurs over a coordinate range $\Delta z = (c_1/c_2)^2$.

For small values of the scalar field at the horizon, *i.e.* as $T \to T_c$, numerical solutions to the equations of motion are found to be well fit by (16) and (17) at the collapse of the ER bridge. See Fig. 3 below. The fitting shows that $c_1/c_2 \approx 0.796(2)\,(1 - T/T_c)^{1/2}$, for numerics with $m^2 = -2$ and $q = 1$. Therefore $\Delta z \to 0$ as $T \to T_c$, so that the collapse becomes very fast in this limit. This justifies the approximation of restricting attention to $z \approx z_\star$. The

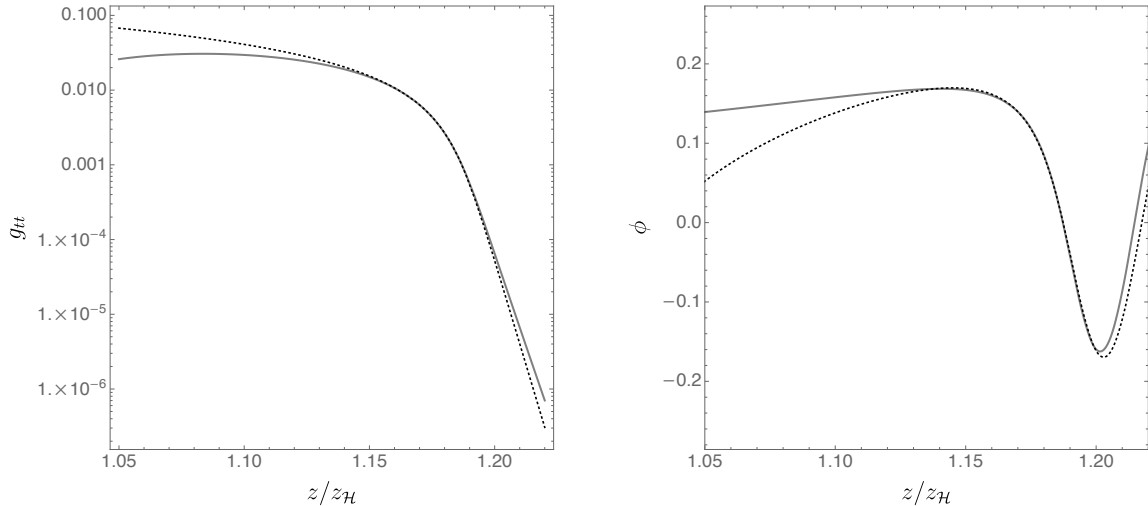

Figure 3: Metric component $g_{tt}$ and scalar $\phi$ as a function of $z$ close to the collapse of the ER bridge. The solid gray line is numerical data and the black dotted curves are fits to the expressions (16) and (17). These are at $T/T_{\rm c} = 0.9936$ and with $m^2 = -2$ and $q = 1$.

vanishing of $c_1/c_2$ as $T \to T_{\rm c}$ is expected because in this limit $z_o, z_\star \to z_{\mathcal{I}}$, the inner horizon of RN AdS, while $E_o, \Phi_o, c_2$ will go to their nonzero values on the RN AdS inner horizon. Therefore in this limit $c_1 \propto \phi_o \to 0$, from (17) and the fact that the scalar field condensate vanishes like $(T_{\rm c} - T)^{1/2}$ as $T \to T_{\rm c}$. While $z_o = z_\star = z_{\mathcal{I}}$ at $T = T_{\rm c}$, the approach to this limit is quite slow. For numerical fitting it is important to keep $z_o$ and $z_\star$ independent.

To verify that it was consistent to neglect the variation in $\Phi$, from the equations above that describe the collapse together with (12) one can show that over the collapse $\Phi' \propto g'_{tt}$. The constant of proportionality is $2/(c_2\sqrt{z_\star[1 - 12/(E_o^2 z_\star^4)]})$ which is order one. Therefore the variation over the collapse $\Delta\Phi \propto \Delta g_{tt} \approx -g_{tt}(z_o)$, as $g_{tt}$ is exponentially small after the collapse. From (17) we have that $g_{tt}(z_o)$ is of order $c_1^2$ up to logarithms. Therefore $g_{tt}(z_o)$ is also small, although not exponentially so, while $\Phi$ is set to a nonzero value by the RN-AdS background. Therefore $\Delta\Phi/\Phi$ is small. Furthermore, we see that $\Phi'$ itself is very small at the end of the collapse, where $g_{tt}$ is decaying exponentially. This fact will simplify the following epoch below.

Finally, it will be useful later to obtain the curvature at the collapse. As a first step, we note that using the solution (17) for the metric, the scalar field (16) can be written as

$$\phi = \phi_o \cos\left(\frac{c_1}{\phi_o c_2}\sqrt{\frac{2}{z_\star}}\log\frac{g_{tt}(z)}{g_{tt}(z_\star)} + \varphi_o\right). \tag{18}$$

Recall that $c_1/\phi_o$ is finite as $\phi_o \to 0$. The Ricci scalar at $z$ can then be written as

$$R = -12 + \frac{q^2 \phi_o^2 \Phi_o^2}{g_{tt}(z)} \cos \left( \frac{2c_1}{\phi_o c_2} \sqrt{\frac{2}{z_\star}} \log \frac{g_{tt}(z)}{g_{tt}(z_\star)} + 2\varphi_o \right) . \tag{19}$$

At fixed small but nonzero $\phi_o$, $g_{tt}$ is exponentially small for $z > z_o$. The curvature is therefore exponentially large. Schematically $R \sim e^{(z-z_o)/\phi_o^2} \cos(z/\phi_o^2)$. The limit $\phi_o \to 0$ is not uniform. While at any nonzero $\phi_o$ there is a large maximum in the curvature right after the collapse, strictly at $\phi_o = 0$ the collapse is simply the smooth inner horizon of AdS RN. We can see this by putting $c_1 = 0$ in (17).

## 4.3 Josephson oscillations

The oscillations in the scalar field in (16) have a physical interpretation. Recall that inside the horizon $z$ is timelike while $t$ is spacelike. The argument of the cosine in (16) can be written as $q \int A_{\hat{t}} d\tau$, where the proper time $d\tau = \sqrt{g_{zz}} dz$ and the vector potential in locally flat coordinates is $A_{\hat{t}} = A_t / \sqrt{g_{tt}}$. A nonzero $A_{\hat{t}}$ indicates a phase winding in the $t$ direction. The scalar condensate $\phi$ determines the superfluid stiffness. Equation (16) therefore describes oscillations in time of the stiffness due to a background phase winding. This is precisely the Josephson effect.[3] After the collapse of the ER bridge, these oscillations become (for $T \approx T_c$) the dominant feature in the solution over a regime that we will now describe.

Immediately after the collapse of the ER bridge described in the previous section, we have seen that $\Phi' \propto e^{-\chi/2}$ is very small. This allows a simplified description of the following regime, wherein we may set $e^{-\chi/2}(E_o^2 - 12/z^4) \to 0$ on the right hand side of (15). Thus

$$\frac{f e^{-\chi/2}}{z^3} = -\frac{1}{c_3}, \qquad \Phi = \Phi_o, \tag{20}$$

with $c_3$ constant. Matching onto the $z > z_o$ side of the collapse, which has an overlapping regime of validity with the oscillation regime, fixes $c_3 = 2(c_2/c_1^2) \times \sqrt{z_\star^5/(E_o^2 z_\star^4 - 12)}$. Therefore this constant becomes large as $T \to T_c$. Using (20), the equations of motion (13) and (15) can then be solved in terms of Bessel functions. The scalar field is given by

$$\phi = c_4 J_0 \left( \frac{|q\Phi_o| c_3}{2z^2} \right) + c_5 Y_0 \left( \frac{|q\Phi_o| c_3}{2z^2} \right) . \tag{21}$$

These Bessel functions are oscillatory and continuously connect onto the oscillations (16) that start in the collapse regime. As $c_3$ is large these oscillations are very fast. Because the

---

[3]Recall that the Josephson current itself has been dropped in (12) because its backreaction on the Maxwell field can be neglected (as we verify below). The current is present in the full Maxwell equation (6).

scalar field oscillations are no longer precisely sinuosoidal, they do backreact onto the metric and we find that in this regime

$$f = -f_o z^3 \exp\left\{ \frac{1}{2} \int_{z_\star}^{z} \left[ \tilde{z}(\phi')^2 + \frac{q^2 \Phi_o^2 c_3^2 \phi^2}{\tilde{z}^5} \right] d\tilde{z} \right\} . \tag{22}$$

Here $f_o$ is a constant of integration. The Bessel functions should be inserted into this integral. The integral can be done analytically in terms of Bessel functions. These describe the small oscillations seen in $zg'_{tt}/g_{tt}$ in Fig. 2 above. At small $T_c - T$, the functional forms (21) and (22) are verified to fit numerical solutions to the full differential equations (6) — (9) all the way from the collapse through to the subsequent Kasner regime. See Fig. 4 below.

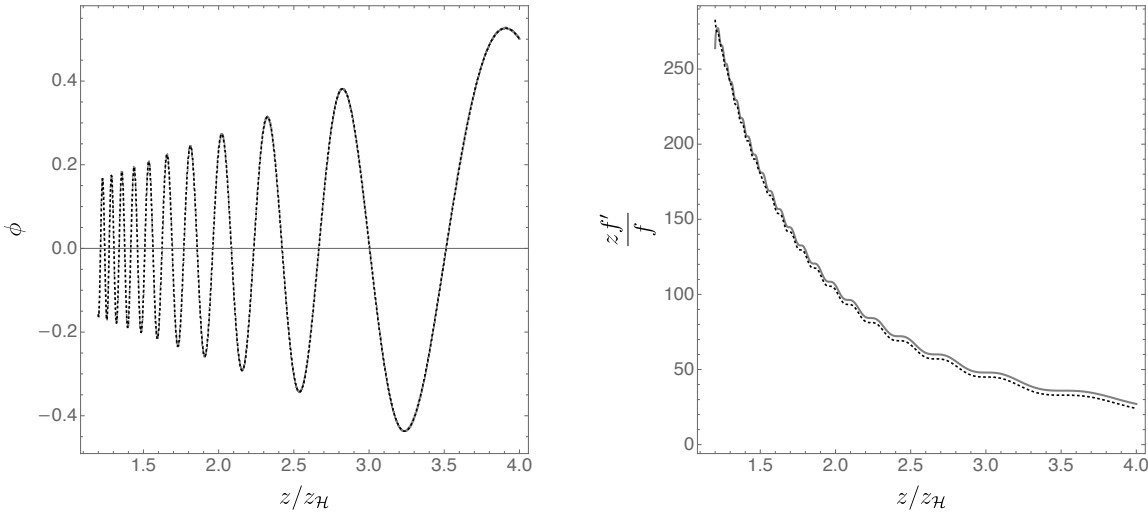

Figure 4: Josephson oscillations in the scalar field $\phi$ and corresponding imprint in the metric derivative $zf'/f$. The solid gray line is numerical data and the black dotted curves are fits to the expressions (21) and (22). These are at $T/T_c = 0.9936$ and with $m^2 = -2$ and $q = 1$.

At large $z$ the scalar field (21) tends to

$$\phi = \frac{2c_5}{\pi} \log\left( \frac{q e^{\gamma_E} \Phi_o c_3}{4z^2} \right) + c_4 + \cdots , \tag{23}$$

with $\gamma_E$ the Euler-Mascheroni constant. The logarithmic behavior indicates the onset of a Kasner regime, that we describe in the following section. The constant $c_5$ in (23) will determine the Kasner exponent. It is therefore interesting to explicitly match this quantity back to the solution at the collapse. Matching (21) and (16) gives

$$c_5 = \left( \frac{z_\star \pi^2 c_2^2}{8} \frac{\phi_o^2}{c_1^2} \right)^{1/4} \sin\left( \frac{c_2 \sqrt{z_\star}}{\sqrt{2}} \frac{1}{\phi_o c_1} - \varphi_o - \frac{\pi}{4} \right) , \tag{24}$$

and similarly for $c_4$. This expression is obtained by matching $\phi$ and its derivative in the regime of overlapping validity and using the expressions for the constants in (17) and below

(20). We have furthermore expanded in the limit of $\phi_o \propto c_1 \to 0$ that applies as $T \to T_c$. The expression (24) shows that $c_5$ is strongly oscillating with constant amplitude as $T \to T_c$:

$$c_5 = A \sin\left(\frac{B}{1 - T/T_c} + C\right). \tag{25}$$

This form agrees with numerics over many oscillations. See Fig. 5 below. For $m^2 = -2$ and

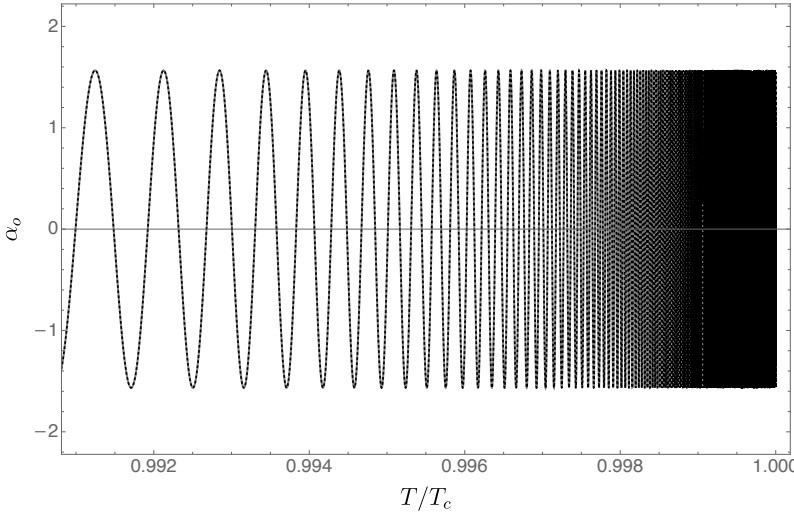

Figure 5: Oscillations in $\alpha_o = -\sqrt{8}c_5/\pi$ (which will determine the subsequent Kasner exponent) as $T \to T_c$. The solid gray line is numerical data and the black dotted curve is a fit to the expression (25). The numerics has $m^2 = -2$ and $q = 1$.

$q = 1$ we find $B \approx 0.491(8)$ and $C \approx 1.945(6)$. The amplitude depends only on the ratio $\xi \equiv z_{\mathcal{I}}/z_{\mathcal{H}}$ of the inner and outer horizon of the RN-AdS background at $T = T_c$:

$$A^4 = \frac{\pi^2 \xi (3 + 2\xi + \xi^2)^2}{16 q^2 (\xi^2 + \xi + 1)}. \tag{26}$$

For a scalar field with $q = 1$ and $m^2 = -2$ this formula predicts $A = 1.7388(6)$, which gives an excellent match to the value we find numerically of $A \approx 1.7393(2)$.

We can again verify that it was self-consistent to treat $\Phi$ as a constant in this regime. Using (20) in the full Maxwell equation (6) gives

$$\Phi' = 2q^2 \Phi_o \frac{z^3}{f} \int \frac{\phi^2}{z^5} dz. \tag{27}$$

Over most of the oscillatory regime, $f$ is exponentially large and hence $\Phi'$ is exponentially small. This allowed the Josephson current to be ignored in the equations of motion. As the Kasner regime is entered, $\Phi'$ is small compared to $\Phi$ by powers of (large) $z$. While the electric flux is negligible upon entering into the Kasner regime, in Sec. 4.5 we will see that it can lead to nontrivial inversion phenomena at large $z$.

## 4.4 Collapse and oscillations at lower temperatures

In Fig. 6 we illustrate how the collapse of the Einstein-Rosen bridge and subsequent Josephson oscillations become less dramatic as the temperature is lowered further below $T_{\rm c}$.

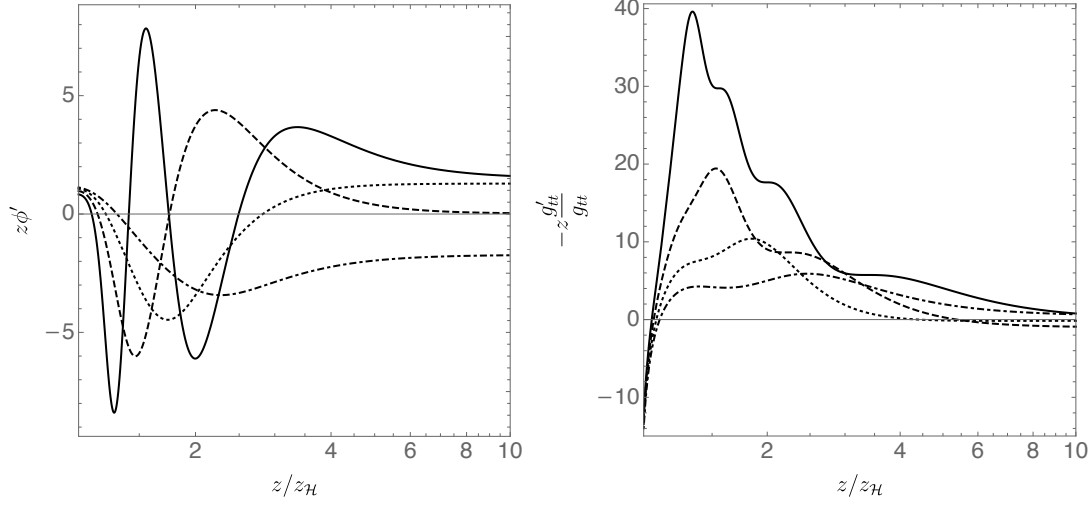

Figure 6: Evolution of $z\phi'$ and $-zg'_{tt}/g_{tt}$ as a function of $z$ for temperatures $T/T_{\rm c} = 0.962$ (solid), 0.934 (dashed), 0.899 (dotted) and 0.858 (dot-dashed). There are fewer oscillations as the temperature is lowered and the jump in the derivative of the metric becomes less dramatic. Numerics are with $m^2 = -2$ and $q = 1$.

## 4.5 Kasner epochs and inversions

We have seen that the oscillatory regime ends in a logarithmic scaling behavior (23). We will shortly explain that this corresponds to a Kasner cosmology. However, this scaling does not always continue all the way to the singularity. Instead, a phenomenon that we call 'Kasner inversion' can occur, as we will also describe.

Beyond the oscillatory regime, the terms involving the charge $q$ of the scalar field can be neglected in (13) and (14) and the cosmological constant term can be neglected in (15). This can be verified a posteriori on the solutions. The resulting equations can be solved in generality (these are just the equations for a massless, neutral scalar field coupled to electromagnetism and gravity without a cosmological constant). Firstly, one solves to find:

$$\Phi = \Phi_o + E_o \int e^{-\chi/2} dz \,, \tag{28}$$

$$\frac{f e^{-\chi/2}}{z^3} = \left( \frac{E_o^2}{4} \int e^{-\chi/2} dz - \frac{1}{c_3} \right) \,. \tag{29}$$

These are the same solutions as in (20), but with $E_o$ reinstated, as it will become important again at larger $z$. This then leaves coupled equations for $\phi$ and $\chi$. We can eliminate $\chi$ from these equations to obtain a third order equation for $\phi$. Setting

$$\phi = \sqrt{2} \int \frac{\psi}{z} dz \, , \tag{30}$$

then the equation becomes

$$\psi^2 + z \frac{\psi''}{\psi'} - 2z \frac{\psi'}{\psi} = 0 \, . \tag{31}$$

The general solution to this equation is

$$(\psi - \alpha_o)^{-1/(1-\alpha_o^2)} \left( \frac{1}{\alpha_o} - \psi \right)^{-1/(1-1/\alpha_o^2)} \psi = \frac{z_{\text{in}}}{z} \, . \tag{32}$$

The two constants of integration are $0 < \alpha_o < 1$ (without loss of generality) and $z_{\text{in}}$.

The solution (32) has the limiting behavior

$$\psi \to \frac{1}{\alpha_o} > 1 \quad \text{as} \quad z \gg z_{\text{in}} \, , \tag{33}$$

$$\psi \to \alpha_o < 1 \quad \text{as} \quad z \ll z_{\text{in}} \, . \tag{34}$$

These limits both describe Kasner cosmologies, as we now explain. Putting $\psi = \alpha$, a constant, into (30) and then solving for the metric variables gives

$$f = -f_K z^{3+\alpha^2} + \cdots , \quad \phi = \alpha\sqrt{2} \log z + \cdots , \quad \chi = 2\alpha^2 \log z + \chi_K + \cdots . \tag{35}$$

Here $f_K, \chi_K$ are constants. The limits in (33) and (34) require $\alpha > 1$ as $z \gg z_{\text{in}}$ and $\alpha < 1$ for $z \ll z_{\text{in}}$. These conditions ensure that the Maxwell flux terms are unimportant in the Kasner regimes where $\Phi = \Phi_K + E_K z^{1-\alpha^2} + \cdots$.

Changing the $z$ coordinate to the proper time $\tau$ (with $\tau = 0$ corresponding to $z = \infty$), using (35) the metric has the Kasner form [10, 15]

$$ds^2 = -d\tau^2 + c_t \tau^{2p_t} dt^2 + c_x \tau^{2p_x} \left( dx^2 + dy^2 \right) , \qquad \phi = -p_\phi \log \tau \, . \tag{36}$$

Here $c_t$ and $c_x$ are constants. The Kasner exponents obey $p_t + 2p_x = 1$ and $p_\phi^2 + p_t^2 + 2p_x^2 = 1$. The single free exponent can be taken to be

$$p_t = \frac{\alpha^2 - 1}{3 + \alpha^2} \, . \tag{37}$$

The sign of $p_t$ determines whether the ER bridge grows ($p_t < 0$) or contracts ($p_t > 0$) as times evolves. In terms of the Kasner exponent $p_t$, the limits in (33) and (34) describe a 'Kasner inversion' $\alpha \to 1/\alpha$ in which

$$p_t \to -\frac{p_t}{2p_t + 1} \, , \tag{38}$$

We see that $p_t$ changes sign in the inversion. Thus $p_t > 0$ at $z \gg z_{\text{in}}$ and $p_t < 0$ for $z \ll z_{\text{in}}$. In particular $p_t > 0$ towards the actual singularity as $z \to \infty$, and hence the $t$ direction contracts at late times. This inversion and late time contraction is clearly seen in Fig. 1.

The late time regime with $p_t > 0$ always exists. However, the intermediate time regime with $p_t < 0$ only exists if the constant $z_{\text{in}}$ in (32) is sufficiently large that $z \ll z_{\text{in}}$ is still within the regime described by (32). If this is not the case, then the full solution can directly enter the late time regime $p_t > 0$ from the oscillating epoch. Matching the end of the oscillatory epoch (23) with the Kasner regime (35) we have $\alpha = -\sqrt{8}c_5/\pi$. Whenever $|\alpha| < 1$ from this matching, there will be a Kasner inversion. The strong oscillations (25) of $c_5$ with temperature mean that $\alpha$ oscillates between $\pm 1.56$ (for $q = 1$ and $m^2 = -2$) infinitely many times as $T \to T_c$. There is therefore an infinite sequence of inverting and non-inverting cosmologies as $T \to T_c$. The following Fig. 7 illustrates an interior evolution that does not exhibit a Kasner inversion.

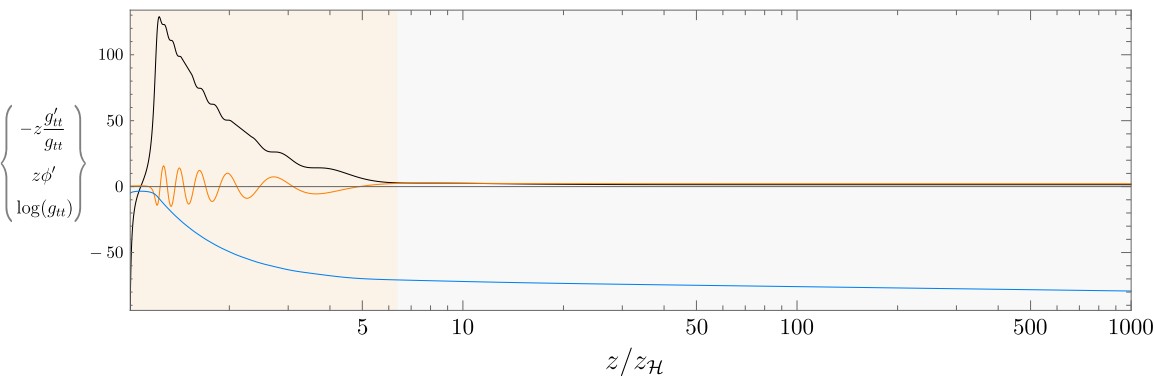

Figure 7: An example of an interior evolution with no Kasner inversion, because the intermediate Kasner exponent is already positive. Numerics with $m^2 = -2$, $q = 1$ and $T/T_c = 0.987$. Compare with Fig. 1, that has a $T/T_c$ differing only by 0.001.

In Fig. 8 we show the value of the Kasner exponent $p_t$ after the inversion for all temperatures up to $T_c$. The strong oscillations near $T_c$ are shown in the blow-up of this region where we plot the values of $p_t$ both before and after the inversion as a function of temperature. These oscillations show an imprint of the Josephson oscillations on the structure of the singularity. Fig. 8 also shows rapid variation of $p_t$ near $T/T_c \sim .6$, illustrated more clearly in a second blow-up plot. This can also be traced back to the Josephson oscillations. Although the number of oscillations tends to decrease with decreasing temperature, as we saw in Fig. 6, it is not monotonic and near $T/T_c \sim .6$ there is a local increase in the number of oscillations. This causes the intermediate $p_t$ to again oscillate below zero, triggering the

inversion.

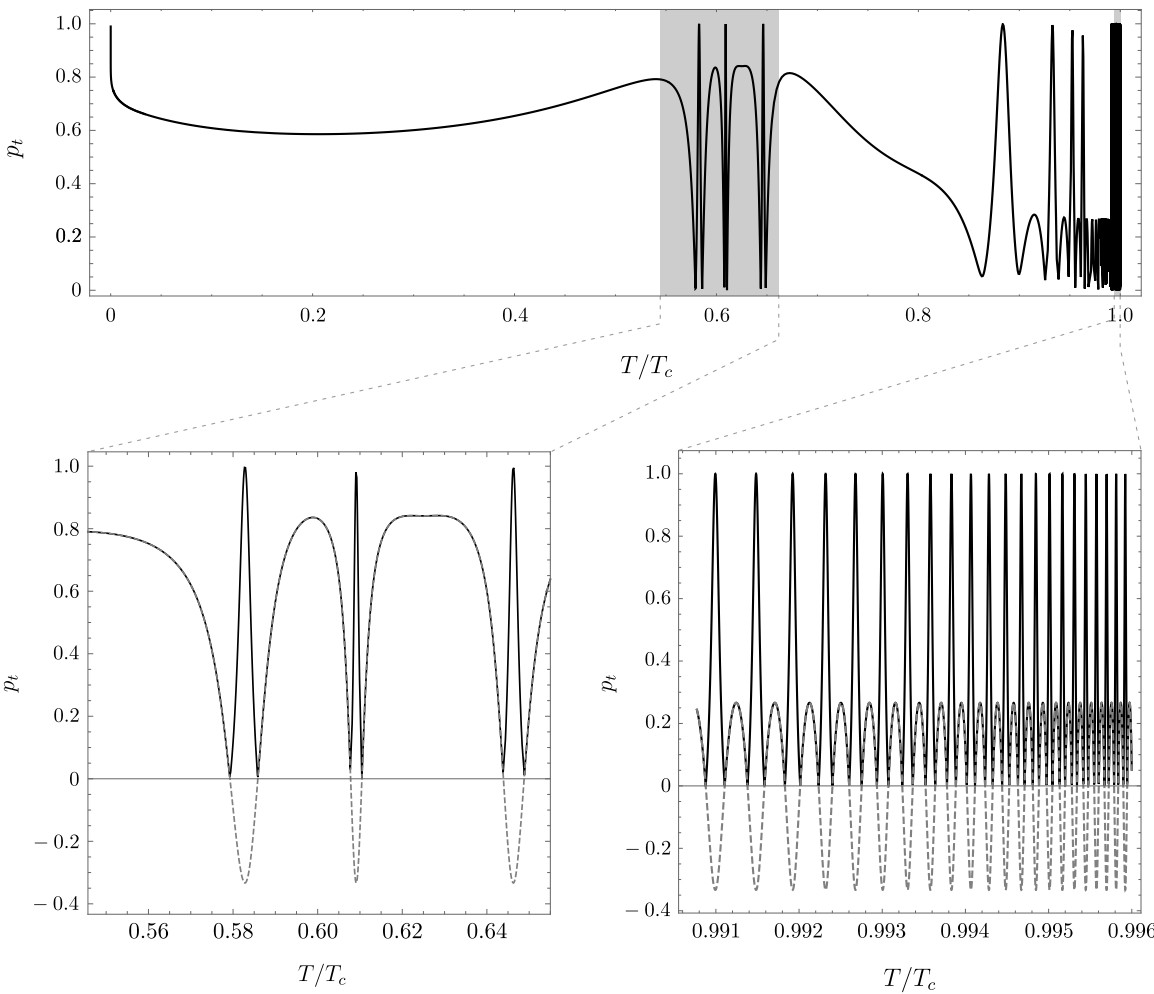

Figure 8: The Kasner exponent $p_t$ after the inversion as a function of $T/T_c$. The oscillating regions have been blown up in the lower two figures. In the lower figures we have also shown the Kasner exponent $p_t^{\text{int}} < 0$ before the inversion as a dashed line. The inversion that occurs when $p_t^{\text{int}} < 0$ is visible. The accumulation of oscillations as $T \to T_c$ is described by (25). Towards zero temperature $p_t \to 1$. We comment more on this in the discussion section.

Numerical solutions to the equations of motion show that the inversions are well described by (32). See Fig. 9 below. As $\alpha_o \to 0$ the inversion becomes increasingly sharp and localized at $z = z_{\text{in}}$. The location $z_{\text{in}}/z_{\mathcal{H}}$ is found (numerically) to tend to a finite number as $\alpha_o \to 0$. This limit is more transparently described in a different coordinate system. The Kasner inversion solution given by (28), (29), (30) and (32) can be written in terms of a different

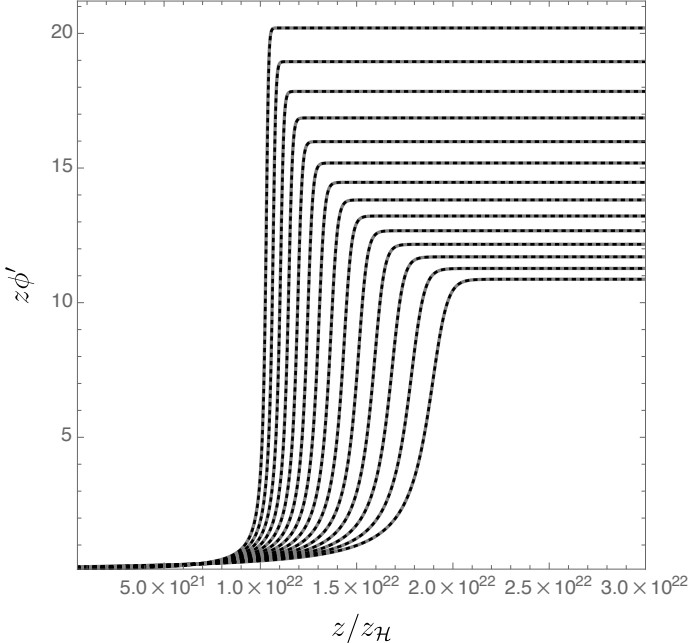

Figure 9: The Kasner inversion in the range $\alpha_o \approx 0.070 - 0.130$, with $T/T_c \approx 0.983$. The solid gray line is numerical data and the black dotted curve is a fit to the expression (32). The numerics are with $m^2 = -2$ and $q = 1$.

'radial' coordinate $r$ as

$$ds^2 = g\,dt^2 - \frac{dr^2}{g} + h\left(dx^2 + dy^2\right)\,, \qquad \phi = \phi_{\rm in} + \sqrt{\frac{1-\beta^2}{2}}\log r\,, \tag{39}$$

where

$$g = g_o^2 \frac{r^\beta}{r_o^\beta \left(r^\beta + r_o^\beta\right)^2}\,, \qquad h = \frac{E_o}{2\beta g_o} r^{1-\beta}\left(r^\beta + r_o^\beta\right)^2\,. \tag{40}$$

The Maxwell field is $d\Phi/dr = E_o/h$. It is easily seen that with $\beta > 0$, the limit $r \gg r_o$ gives a Kasner solution with $p_t = -\beta/(2+\beta) < 0$. This corresponds to the $z \ll z_{\rm in}$ limit discussed above. The opposite limit of $r \ll r_o$ gives a Kasner solution with $p_t = \beta/(2-\beta) > 0$. Thus we precisely recover the Kasner inversion (38) in these new coordinates. The fact that the scalar in (39) takes the same form in both asymptotic regions is still consistent with (32) since the coordinate transformation from $z$ to $r$ is different in the two regions.

The benefit of the new coordinates is that it is straightforward to take the limit $\beta \to 1$, which corresponds to $\alpha_o \to 0$. The solution becomes

$$ds^2 = \frac{\hat{r}}{\hat{r}_o(\hat{r}+\hat{r}_o)^2}d\hat{t}^2 + (\hat{r}+\hat{r}_o)^2\left(-\frac{\hat{r}_o}{\hat{r}}d\hat{r}^2 + d\hat{x}^2 + d\hat{y}^2\right)\,, \qquad \phi = \phi_{\rm in}\,. \tag{41}$$

We introduced rescaled coordinates $\hat{r}, \hat{t}, \hat{x}, \hat{y}$ for clarity (*i.e.* to remove $g_o$ and $E_o$). The Maxwell potential is now $\Phi = \Phi_o - 2/(\hat{r} + \hat{r}_o)$. This solution describes a crossover from

$p_t = -1/3$ at large $\hat{r}$, to $p_t = 1$ at small $\hat{r}$. The solution (41) has a constant scalar field and is in fact a special case of a class of known exact solutions in Einstein-Maxwell theory [16] that can also exhibit Kasner inversion.[4]

As $\hat{r} \ll \hat{r}_o$ in (41) the geometry tends towards a Kasner solution with $p_t = 1$. This exponent corresponds to a regular horizon, and would therefore lead to a smooth inner horizon of our spacetime. The theorem in section 3 precludes this possibility. Therefore, in this particular limit of $\alpha_o \to 0$, the terms involving the charge of the scalar field — that are otherwise negligible at large $z$ — must become important at the inversion and prevent $\alpha_o = 0$ from inverting to a new value of $\alpha_{\text{new}} = \infty$ (which corresponds to $p_t = 1$). The simplest scenario for what occurs when charge terms are important is that there is a repeat of the Einstein-Rosen bridge described in section 4.2. Using results from that section, we can verify that the charge terms indeed render $\alpha_{\text{new}}$ finite as $\alpha_o \to 0$. Expression (24) gives the value of $\alpha_{\text{new}} = -\sqrt{8}/\pi \times c_5$ after the inversion/collapse, in the presence of charge terms. (Previously $c_5$ was related to $\alpha_o$, but now we are applying this formula to a second collapse). Expressed in terms of quantities that remain finite and nonzero at the collapse — using (17) and dropping the $-12$ term that is unimportant at large $z_{\text{in}}$ — from (24) we have that the magnitude of $\alpha_{\text{new}}$ is bounded

$$|\alpha_{\text{new}}|^2 \leq \alpha_{\text{max}}^2 \equiv \frac{2}{\pi} \frac{c_2 E_{\text{in}} z_{\text{in}}^{5/2}}{q \Phi_{\text{in}}} . \tag{42}$$

Here $\Phi_{\text{in}}$ and $E_{\text{in}}$ are the values of the potential and electric field at the inversion. At any nonzero $q$, therefore, $\alpha_{\text{new}}$ cannot diverge even if $\alpha_o \to 0$. The new Kasner exponent is therefore strictly less than one.

Fig. 10 gives an illustration of the bound (42) in action. The plots show the evolution of the Kasner inversion/'second collapse' as $\alpha_o$ is tuned through zero, and the corresponding behavior of $\alpha_{\text{new}}$. In Fig. 10 we furthermore see the appearance of an oscillation at the inversion, due to the charge terms. These oscillations are to be expected once the inversion is described by the equations in section 4.2. In fact, such oscillations are necessary to interpolate between solutions with large positive and negative $\alpha_{\text{new}}$ that necessarily arise from the inversions of $\alpha_o$ small and positive and $\alpha_o$ small and negative, on either side of the developing zero in $\alpha_o$. In general it is difficult to see these oscillations numerically at inversions where

---

[4]In general $ds^2 = g^2(\tau)\left[-d\tau^2 + \tau^{2p_x}dx^2 + \tau^{2p_y}dy^2\right] + g(\tau)^{-2}\tau^{2p_t}dt^2$ and $A_t = (2k_2/k_1)^{1/2}/g(\tau) + \Phi_0$, with $g(\tau) = k_2 + k_1\tau^{2p_t}$. The only constraint on the two constants $k_1, k_2$ is that their product is positive. The exponents must satisfy the usual constraints $p_t + p_x + p_y = 1$ and $p_t^2 + p_x^2 + p_y^2 = 1$. If $p_t > 0$, $g \to k_2$ as $\tau \to 0$ and the Kasner exponents do not change. But if $p_t < 0$, $g$ diverges in this limit. The metric near $\tau = 0$ is again of the Kasner form but now with $p_t$ replaced with $-p_t/(2p_t + 1)$, exactly as in (38).

$z_{\mathrm{in}}$ is typically significantly larger than in Fig. 10. That is because the bound in Eq. (42) is saturated at very large $\alpha_{\mathrm{new}}$ in those cases, requiring the system to get to very small $\alpha_o$ before the effects of charge becomes important. That is, extremely precise tuning in $T/T_{\mathrm{c}}$ is needed.

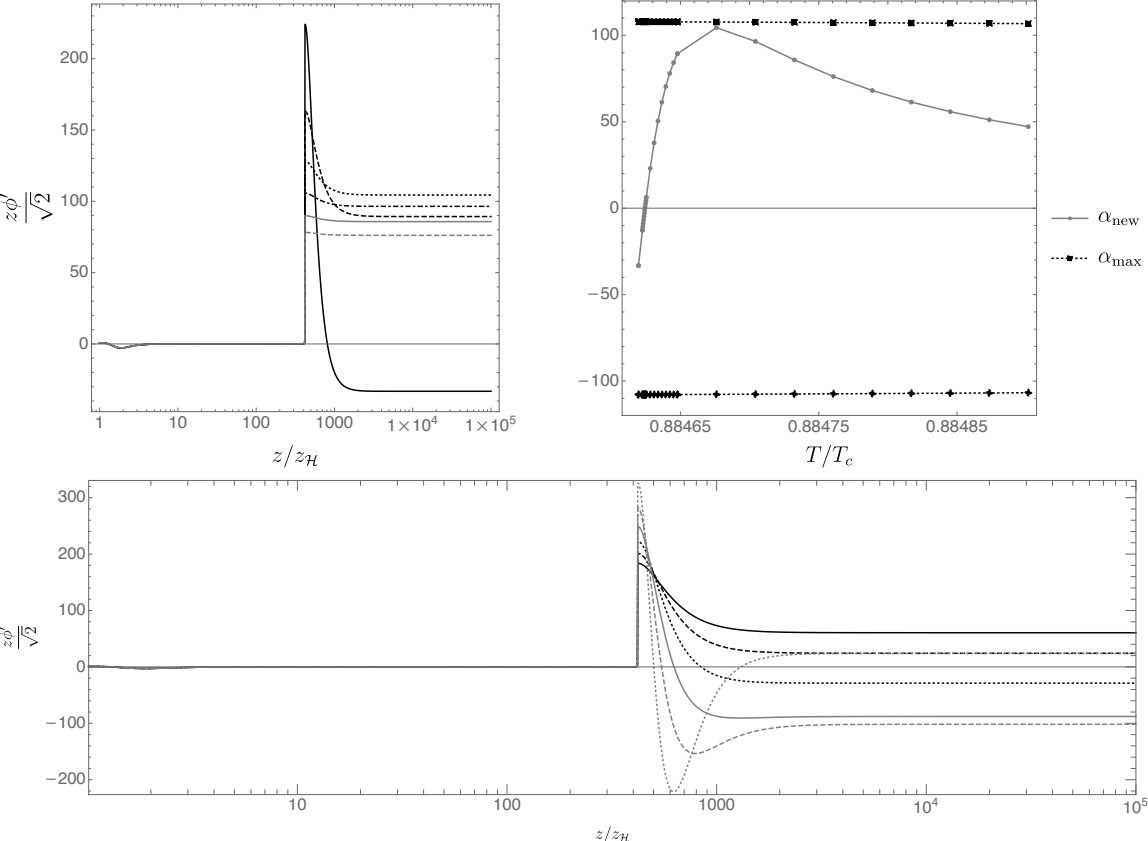

Figure 10: Approach to the bound (42) on $\alpha_{\mathrm{new}}$ (top) and appearance of an oscillation at a Kasner inversion (bottom). In the upper left we plotted for temperatures $T/T_c = 0.88462$ (solid black), 0.88465 (dashed black), 0.88468 (dotted black), 0.88470 (dot-dash black), 0.88473 (solid gray) and 0.88476 (dashed gray), showing the decrease in $z\phi'$ after the inversion as the bound is approach. The lower figure has $T/T_c = 0.884684$ (solid black), 0.884676 (dashed black), 0.884669 (dotted black), 0.884661 (solid gray), 0.884652 (dashed gray) and 0.884644 (dotted gray) in order to highlight the eventual emergence of oscillations after the second collapse. Notice the extreme sensitivity of the new value $\alpha_{\mathrm{new}}$ on the temperature.

The appearance of additional oscillations at all inversions once $\alpha_o$ gets sufficiently small has a fascinating consequence for the structure of interior solutions. As we see in the bottom plot in Fig. 10, the oscillation leads to the development of a zero in $\alpha_{\mathrm{new}}$. We know that Kasner solutions with $|\alpha_{\mathrm{new}}| < 1$ cannot continue asymptotically and so there must be a

second inversion in this case. As $\alpha_{\text{new}} \to 0$ this second inversion must again become very steep and ultimately lead to oscillations. These oscillations in turn seed further inversions. It is clear that we are led to an infinite sequence of Kasner regimes and oscillatory Kasner inversions. The number of distinct Kasner regimes would depend sensitively on $T/T_c$ in a fractal-like way, with additional very fine oscillations in temperature arising within the structure of (25), due to these inversions.

It is numerically extremely challenging to see these further inversions, but for $T/T_c \approx$ 0.8846244786, which has $\alpha_{\text{new}} \approx 0.72$ at the end of the first inversion, we see evidence of a second inversion developing at $z_{\text{in}(2)} \approx 1.71 \times 10^{20322}$.

## 5   Discussion

We have studied what happens inside the horizon of the simplest holographic supercon-ductor. We showed that the Cauchy horizon present for temperatures above the critical temperature is always removed when $T < T_{\text{c}}$ and replaced by a spacelike singularity. For $T$ close to $T_{\text{c}}$, the interior dynamics cleanly separates into different epochs which can be described as the collapse of the Einstein-Rosen bridge, Josephson oscillations, and a Kasner regime. In some cases, there can be a sequence of transitions between Kasner regimes as the singularity is approached. At certain temperatures these transitions become very abrupt, re-enacting the collapse of the Einstein-Rosen bridge, seeding a new set of Josephson oscilla-tions and leading to a remarkable recursive structure as a function of $z$ (for a given solution) and fractal structure as a function of $T/T_{\text{c}}$ (the space of solutions).

Even a single abrupt collapse of the Einstein-Rosen bridge, at the would-be Cauchy horizon as $T \to T_{\text{c}}$, is sufficient to lead to an extreme sensitivity of the final Kasner exponent on the temperature near $T_c$. This was illustrated in Fig. 8. For any $\epsilon > 0$, the Kasner exponent cycles through a finite range of values *an infinite number of times* as the temperature is lowered from $T_c$ to $T_c - \epsilon$. The accumulation of oscillations in the final Kasner exponent as $T \to T_{\text{c}}$ is due to an accumulation of oscillations in the scalar field just beyond the (now absent) inner horizon of AdS RN. The destruction of the inner horizon becomes increasingly sudden as the condensate vanishes as $T \to T_{\text{c}}$, leading to a divergent curvature just beyond the inner horizon that we computed in (19). Eventually, then, string theoretic or quantum gravity effects will be important at the inner horizon, potentially cutting off the infinite oscillations in temperature.

In Fig. 8 each solution (except for those with $p_t$ very close to one) approaches a fixed

Kasner epoch near the singularity. The infinite number of Kasner cycles arises by tuning an external parameter — the temperature — rather than by time evolution. However, we have also seen that these oscillations in temperature are not the whole story. The analytic result (25) determines the oscillatory Kasner exponent after a first collapse of the Einstein-Rosen bridge. However, this Kasner exponent is then subject to Kasner inversions. At certain temperatures, these inversions can become sudden and seed further oscillations in such a way that the whole process repeats itself an infinite number of times. This infinite repetition occurs at a discrete set of temperatures $\{T_n\}$ close to where $\alpha_o = 0$ in Fig. 5, or $p_t = 1$ in Fig. 8. These temperatures $T_n$ accumulate at $T_c$. We find it remarkable that the onset of the scalar outside the horizon is associated with the most intricate dynamics inside. The chaotic sequence of Kasner epochs — now as function of $z$ — is rather reminiscent of the mixmaster singularity [11,17]. As noted in the introduction, in contrast to the usual BKL analysis, the oscillations due to the charge of the scalar field are essential for the dynamics in our case. In terms of temperature dependence, this phenomenon is expected to lead to a very fine fractal structure superimposing itself on Fig. 8. It would be extremely interesting to exhibit this structure explicitly in future work, with either a more refined numerical approach or more powerful mathematical tools.

Further interesting questions for future work include the extent to which our results are modified in theories with more general scalar potentials. The collapse and inversions in particular are nonlinear regimes that are likely sensitive to the potential. There are also models of inhomogeneous holographic superconductors [18]. The fact that different spatial points decouple near a spacelike singularity suggests that the behavior near the singularity in those cases might be similar to the homogeneous solutions. It would be interesting to see if that is the case. In some circumstances boundary time dependence should also translate into spatial dependence in the interior that can be expected to exhibit pointwise decoupling near the singularity.

In [9] we noted that the collapse of the ER bridge was associated with a critical interior radius $z_c$ where $g'_{tt}(z_c) = 0$. Such interior extrema can be associated to purely damped quasinormal modes (in the exterior) for fields with large mass $M$, with frequency $\omega = -iM\sqrt{g_{tt}(z_c)}$ [6]. We explicitly verified the existence of the corresponding quasinormal mode in that case. In the present case of a gravitating charged scalar field each Kasner inversion comes with an additional maxima of $g_{tt}$. Could these lead to a plethora of overdamped modes? In general, whether an extremum contributes to the late time decay of fields depends on analytic properties of the geodesics in the complex energy plane, which we do

not have access to here given our numerical backgrounds [19, 20]. From explicit studies of the quasinormal mode spectrum, we have only been able to identify a quasinormal mode associated to the first collapse. This suggests that the other potential modes do not exist. We have also checked that higher dimensional surfaces that traverse the ER bridge and capture entanglement in the dual field theory [6] do not have additional extrema associated with the Kasner inversions. We hope that the intricate classical dynamics that we have found inside the horizon will motivate further attempts to probe this region from the dual theory.

The fractal-like structure described above does not affect the global causal structure of our solutions. The causal structure of AdS black holes without Cauchy horizons can be represented by a Penrose diagram where it makes a difference whether the spacelike singularity bends in toward the event horizon or out away from it. For $T$ close to $T_c$, the singularity in the holographic superconductor bends away (*i.e.* upwards in the Penrose diagram) and is close to the former Cauchy horizon. At intermediate temperatures the singularity comes down, eventually changing convexity and approaching the event horizon as $T \to 0$. This is consistent with the $p_t \to 1$ behavior seen in Fig. 8, although different from the zero temperature limit for neutral scalar fields discussed in [9]. The singular horizon at $T = 0$ was found in [21].

## Acknowledgments

We thank Rong-Gen Cai, Li Li, and Run-Qiu Yang for pointing out a problem with an argument for inhomogeneous horizons that we had in the first version of this paper. S. A. H. is supported by DOE award DE-SC0018134 and by a Simons Investigator award. G. H. is supported in part by NSF grant PHY1801805. J. K. is supported by the Simons foundation. J. E. S. is supported in part by STFC grants PHY-1504541 and ST/P000681/1. J. E. S. also acknowledges support from a J. Robert Oppenheimer Visiting Professorship. S. A. H. acknowledges the hospitality of the Max Planck Institute CPfS while this work was being finalized. This work used the DIRAC Shared Memory Processing system at the University of Cambridge, operated by the COSMOS Project at the Department of Applied Mathematics and Theoretical Physics on behalf of the STFC DiRAC HPC Facility (www.dirac.ac.uk). This equipment was funded by BIS National E- infrastructure capital grant ST/J005673/1, STFC capital grant ST/H008586/1, and STFC DiRAC Operations grant ST/K00333X/1. DiRAC is part of the National e-Infrastructure.

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
