# Peer review of "Diving into a holographic superconductor"

_SciPost Physics_

## Round 2 · Referee Report · Anonymous (Referee 1) · 2020-10-2

Strengths

See report

Weaknesses

See report

Report

This paper fills an important gap in the holographic realization of superconductivity, by systematically studying the dynamics of the relevant fields in the dual black hole interior. The authors show that the charged scalar field eliminates the inner Cauchy horizon. Hence the singularity inside the black hole is spacelike and a BKL-like analysis applies.

The analysis of the Einstein-Maxwell-scalar system near a spacelike singularity by BKL techniques has a long history, and the dynamical features near the spacelike singularity uncovered by the authors confirms to some extent old results (see note below). But the holographic understanding is certainly original and extremely interesting, so that I recommend publication.

I would just suggest that the authors read and perhaps take into account the information in the note below. Whatever they choose to do, however, I think that the paper can be accepted and I do not need to see the manuscript again.

Note: Rigorous analytical results on the “subcritical” Einstein-Maxwell-scalar system can be found in Commun.Math.Phys. 218 (2001) 479-511 (gr-qc/0001047) and Annales Henri Poincare 3 (2002) 1049-1111 (gr-qc/0202069). Information on the generic approach to a final Kasner regime was given earlier in Phys. Lett. B175 ( 1986) 129-132 (being “generic”, the approach does not exclude an infinite number of oscillations (“mixmaster type”) for conditions of “zero measure”).

Requested changes

See report

---

## Round 2 · Referee Report · Anonymous (Referee 2) · 2020-10-24

Report

The manuscript “Diving into a holographic superconductor” by S. Hartnoll, et.al. studies the geometry of black hole interior for classical AdS Einstein-Maxwell-charged scalar gravity at low temperatures $T<T_c$ when a nonzero scalar field is favored. Using both analytic analysis and numerical calculations they found that no inner horizon exists and along the timelike coordinate inside the black hole, the geometry would pass through different intermediate regimes, including “collapse of Einstein-Rosen bridge”, “Josephson oscillations”, the Kasner universe and finally reaches a spacelike Kasner singularity.

The geometry inside the black hole horizon is currently being paid attention to, which might be related to quantum information of quantum fields outside the black hole horizon. This work provides an intuitive example of the black hole interior in the classical regime. The physics described in the manuscript is clear and the picture is easy to follow by the readers. The analytic analysis is further confirmed by numerics which should be sound and trustable. I am only a little confused by certain analytic arguments about what terms could be ignored and what terms should be important. I list several of the confusing arguments as examples in the following. 1) In the second paragraph of section 4.2, $\delta z$ dependence in functions of $f$, $\chi$, etc. is kept but in the equations is not kept, is it possible to have a simple explanation why $\delta z$ dependence in the parameters of the equations could be ignored and why $\delta z$ dependence in the functions have to be kept? 2) Under (17), what is the value of $g_{tt}$ at $z=z_0$? 3) On the top of page 10, it is argued that $\Phi’$ could be ignored because $\delta \Phi \to 0$ as $\delta z \to 0$, however we need $\delta \phi/\delta z\to 0$ for $\Phi’$ to be ignored. 4) In the first paragraph of section 4.2, why is this called the “collapse of the Einstein-Rosen bridge”?

I do not doubt that these arguments should be correct as they have been further confirmed by numerics. Thus I suggest the authors elaborate a little more on the explanations for these arguments and make them more clear and detailed. I recommend the manuscript to be published after these modifications to the text.

---

## Round 2 · Referee Report · Anonymous (Referee 3) · 2020-10-27

Strengths

see report

Weaknesses

see report

Report

This manuscript studies the interior of planar AdS black hole after the complex scalar field begins to condensate. It finds there is no inner horizon. It also observes the Josephson oscillations and the Kasner behavior of singularity according to numerical computations and semi-analytical analyses.

Basically, I think results in this manuscript are interesting and should be published but I feel there are still a few of aspects that should be improved. I call the analytical arguments of this manuscript to be`` semi-analytical analyses’’ because the authors made a few of addition assumptions without enough arguments or proofs. The authors used "posteriori’’ to support the validity of these assumptions. However, I feel this has suspicion of circularity.

In the Sec.4.1, authors claimed that the mass terms of scalar field and charged terms of Maxwell term are neglectable when $T$ is closed to $T_c$, then they find that Eqs (6)-(9) are simplified into Eqs.(12)-(15). It is clear that the mass term in Eq (9) is neglectable as scalar field is small. However, at least for me, it is completely difficult to understand why we can do that in other three equations.

For example, the Eq. (7) is only a linear equation of $\phi$, I do not understand why the mass can be dropped. Particularly, from Eq. (6) it seems that $\Phi/f$ approaches to finite when we come close to the would-be inner horizon. Thus, the $\Phi^2/f$ term will approach to zero. If these is true, we would obtain a conclusion that, in Eq. (13), the $\Phi$ term should be dropped but the mass term should be kept.

The similar puzzlement also appears when the authors obtained Eqs. (28) and (29).

There is also another place which confuses me. In Sec. 4.5, authors use equation (12)- (15) to obtain Eq. (28) and (29). Then they discussed lower temperature case, i.e. $T\ll T_c$. But in Sec. 4.1, authors said “At temperatures close to Tc…. At that point there are…”. If my understanding is correct, the equations (12)-(15) is obtained in the case when $T$ is closed to $T_c$. Why can they be used when $T$ is much smaller than $T_c$?

In addition, between Eqs. (35) and (37), I found, if I do not make mistake,
$p_x=2/(3+\alpha^2)$ and $p_\phi=2\alpha/(3+\alpha^2)$
Then $p_\phi^2+p_t^2+2p_x^2\neq1$.
A typo may be in the part between Eqs. (35) and (37).

To summary, I suggest the authors try to give more arguments and explanations on their assumptions of analytical analyses. I would be happy to recommend the publication after above issues are responded suitably in their revised version.

---

## Editorial Decision

resubmitted